# Usability Evaluation—Advances in Experimental Design in the Context of Automated Driving Human–Machine Interfaces

**Deike Albers** [1],*[ID]**, Jonas Radlmayr** [1]**, Alexandra Loew** [1]**, Sebastian Hergeth** [2][ID]**,
Frederik Naujoks** [2][ID]**, Andreas Keinath** [2] **and Klaus Bengler** [1]

[1]  Chair of Ergonomics, Technical University of Munich, Boltzmannstraße 15, 85748 Garching, Germany;
    jonas.radlmayr@gmail.com (J.R.); alexandra.loew@tum.de (A.L.); bengler@tum.de (K.B.)
[2]  BMW Group, Knorrstraße 147, 80937 Munich, Germany; Sebastian.Hergeth@bmw.de (S.H.);
    Frederik.Naujoks@bmw.de (F.N.); andreas.keinath@bmw.de (A.K.)
*   Correspondence: deike.albers@tum.de; Tel.: +49-89-289-15420

**Abstract:** The projected introduction of conditional automated driving systems to the market has sparked multifaceted research on human–machine interfaces (HMIs) for such systems. By moderating the roles of the human driver and the driving automation system, the HMI is indispensable in avoiding side effects of automation such as mode confusion, misuse, and disuse. In addition to safety aspects, the usability of HMIs plays a vital role in improving the trust and acceptance of the automated driving system. This paper aggregates common research methods and findings based on an extensive literature review. Empirical studies, frameworks, and review articles are included. Findings and conclusions are presented with a focus on study characteristics such as test cases, dependent variables, testing environments, or participant samples. These methods and findings are discussed critically, taking into consideration requirements for usability assessments of HMIs in the context of conditional automated driving. The paper concludes with a derivation of recommended study characteristics framing best practice advice for the design of experiments. The advised selection of scenarios and metrics will be applied in a future validation study series comprising a driving simulator experiment and three real driving experiments on test tracks in Germany, the USA, and Japan.

**Keywords:** conditionally automated driving; human–machine interface; usability; validity; method development

---

## 1. Introduction

The introduction of conditionally automated driving (CAD) vehicles drastically alters the role of the human in the car. Based on the definition of the Society of Automotive Engineers (SAE), CAD or Level 3 automated driving means that the automated driving system (ADS) is responsible for the entire driving task, while the human operator is ready to respond as necessary to ADS-issued requests to intervene and to system failures by resuming the driving task [1]. The transition of the human driver from the role of operator to the passenger role implies a paradigm change relative to the Level 2 or partially automated systems that are available today [1,2]. This paradigm change, including transitions back and forth to lower levels of automated driving, affects the human–machine interface. CAD implies that the human must take back control of the driving task in cases where the system reaches a system boundary and in doing so, to resume manual driving. The resulting transition of the driving task from the automation system to the human requires an appropriate communication strategy as well as a human–machine interface (HMI) that supports the interaction between the two

parties in general. New challenges in both HMI design for automated driving and CAD in particular are addressed in this review paper.

This paper gives an overview of the status quo for usability assessments for automated driving HMIs. Current practice is presented by summarizing the methodological approaches of study articles. Additionally, theoretical articles such as literature reviews are included. Both are considered in the derivation of best practice advice for experimental design. This best practice advice will be applied in an international validation study for assessing the usability of CAD HMIs comprising four experiments in three countries and two testing environments. In Germany, a driving simulator experiment and a test track experiment are planned. Two further test track experiments are planned for Japan and the USA. All four experiments will apply the same study design, ensuring the comparability of the results. The articles in this paper have been aggregated using a predefined set of six categories. These categories were identified in the research phase of the validation project and represent differences in the methodological approaches.

Basing on the existing literature, this paper aims to derive a feasible practical and theoretical experimental design that will be validated in the study series described above. The developed experimental design serves as best practice for future studies in which the aim is to assess the usability of CAD HMIs.

## 2. Paper Selection and Aggregation

This paper reviews 16 scientific articles that cover the usability assessment of CAD HMIs. The selection includes study articles and theoretical articles. The selection process and the aggregated data are presented in the following sections.

### 2.1. Paper Selection

Literature searches have been conducted in the literature manager Mendeley and Researchgate, resulting in seven articles.

Additionally, a systematic review has been conducted via the search engine for scientific literature Google Scholar. The process followed the guideline Preferred Reporting Items for Systematic Reviews and Meta-Analyses (PRISMA) and is visualized in Figure 1 [3]. This guideline enhances the transparency of the selection process by describing the stepwise narrowing of the chosen articles for the review. For the identification of potential articles, different combinations of keywords such as "Usability", "Human–Machine Interface", and "Conditionally Automated Driving" are applied. The first step in the process resulted in 553 articles. The next step included the articles identified in other libraries or databases, respectively. In total, 188 duplicates were removed. A first screening of the titles and the abstracts lead to the exclusion of 346 further articles. After reading all articles, 10 more articles were excluded due to a lack of relevance for this review. In the resulting selection of 16 articles, the usability assessment of ADS HMIs for CAD is described.

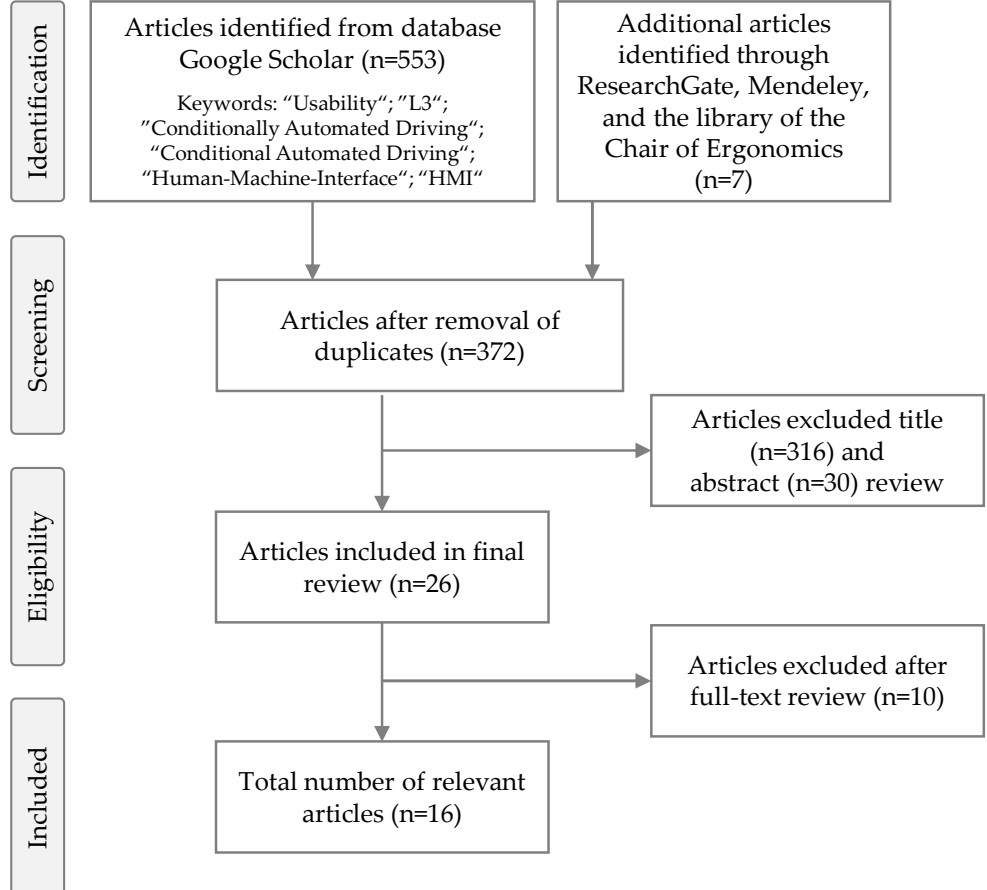

**Figure 1.** Process of the literature review based on the PRISMA guideline [3].

*2.2. Aggregation*

The final selection includes 16 articles. Nine articles present experiments, which are hereafter referred to as study articles [4–12]. The seven other articles are of theoretical nature and are therefore referred to as theoretical articles [13–19]. There are several characteristics that define a study design. By taking into account both common practice and theoretical considerations, this review paper aims to derive best practice advice for researchers interested in the usability of CAD HMIs.

Six experiment characteristics were chosen to meet the challenge of assessing usability in the development process. The literature search yielded different approaches for the usability testing of ADS HMIs. The differences identified in the first research phase resulted in the selection of six categories. These provide the structure of this paper, including the study characteristics' dependent variables, and the testing environment. The definitions of the term usability applied in each of the selected articles are used to understand the research focus of each article. Furthermore, the sample characteristics, the test cases, and the conditions of use, i.e., initial versus repeat contact (see below), are considered. The characteristics listed below provide an insight into the methodological approaches of the nine empirical study articles and the discussed and recommended methodologies of the seven theoretical articles:

- Definition of Usability
- Testing Environment
- Sample Characteristics
- Test Cases
- Dependent Variables
- Conditions of Use

The characteristics listed or applied in the 16 articles are summarized in the first paragraph of the following subsections and the respective tables. Every subsection closes with a critical discussion of the findings resulting in a recommendation of an experimental procedure or method. These recommendations form the best practice advice for usability assessments of CAD HMIs.

### 2.2.1. Definition of Usability

The understanding of the term usability has a considerable influence on the experimental design that researchers choose. Different definitions and operationalizations may result in a different study design. To reflect these potential differences in design, the information on usability given in the selected articles is compared in this subsection. Table 1 shows 12 of the 16 articles. Four articles do not define or operationalize the term usability [4,5,8,15]. Five of the remaining articles [9,11,13,18,19] give an insight into the authors' understanding of the construct usability by the chosen dependent variable(s), e.g., the acceptance, or metrics, e.g., the System Usability Scale (SUS) [20], the Post-Study System Usability Questionnaire (PSSUQ) [21], or the acceptance scale of Van der Laan (VDL) [22]. Four articles [6,7,12,17] cite ISO Standard 9241 with its definition of usability as the "extent to which a system, product or service can be used by specified users to achieve specified goals with effectiveness, efficiency and satisfaction in a specified context of use" [23] (p. 2). However, the complete definition is used only once [7], whereas three articles focus on the effectiveness and efficiency while leaving out the construct satisfaction [6,12,17]. Ref. [12] adds the term "usefulness" to the constructs effectiveness and efficiency. Two other articles cite the minimum requirements of the National Highway Traffic Safety Administration (NHTSA) [16,17]. These requirements impose that the user of an ADS HMI must be able to understand if the ADS is "(1) functioning properly; (2) currently engaged in ADS mode; (3) currently 'unavailable' for use; (4) experiencing a malfunction; and/or (5) requesting control transition from the ADS to the operator" [24] (p. 10). Ref. [17] applies the NHTSA minimum requirements to the two constructs effectiveness and efficiency. The remaining two articles [10,14] cite Nielsen [25] who builds usability from five constructs: learnability, efficiency, memorability, errors, and satisfaction.

**Table 1.** Aggregation of the definitions of usability.

| Article | ISO Standard 9241 [23] | Nielsen [25] | NHTSA Minimum Requirements [24] | Operationalization Through Dependent Variables |
|---|---|---|---|---|
| Forster et al. (2019c) [6] | Effectiveness and efficiency | | | |
| Forster et al. (2019d) [7] | x | | | |
| Kettwich et al. (2016) [9] | | | | Satisfaction and usefulness (VDL [22]), expectations, suggestions |
| Morgan et al. (2018) [10] | | x | | |
| Naujoks et al. (2017) [11] | | | | Comprehensibility, SUS [20] |
| Richardson et al. (2018) [12] | Efficiency, effectiveness, and usefulness | | | |
| Forster et al. (2018) [1] [13] | | | | SUS [20]/PSSUQ [21] |
| François et al. (2016) [1] [14] | | x | | |
| Naujoks et al. (2018) [1] [16] | | | Usability and safety | |
| Naujoks et al. (2019a) [1] [17] | Effectiveness and efficiency | | x | |
| Naujoks et al. (2019b) [1] [18] | | | | 20-item guideline |
| Pauzie and Orfila (2016) [1] [19] | | | | Acceptability, acceptance, trust, situation awareness, workload |

[1] Theoretical article.

In addition to the implicit operationalization through dependent variables, only three sources are cited for the 16 selected articles. These are the ISO Standard 9241 [23], the NHTSA minimum requirements [24], and the Nielsen model for usability [25]. During examination of the articles, both theoretical articles and study articles posed difficulties in working out the authors' understanding of usability. Considering that usability forms the focus of the research question, the underlying definition or at least the operationalization should be communicated to the readers. We strongly advice applying ISO Standard 9241 that comprises the constructs effectiveness, efficiency, and satisfaction [23]. Since the ISO Standard does not elaborate on the detailed testing procedure, further operationalizations are recommended, e.g., whether the effectiveness is tested in a setting with novice or experienced users. When citing the NHTSA requirements for usability tests, researchers choose a different approach to defining the term usability that considers the context of automated driving. Moreover, the usability is rated according to the comprehension of the user that the ADS is "(1) functioning properly, (2) currently engaged in ADS mode, ... " [24] (p. 10). This narrows down the practical realization of the usability assessment. A combination of this approach and the definition of the general term usability based on ISO Standard 9241 seems to be most applicable.

### 2.2.2. Testing Environment

Four of the theoretical articles provide no information on the testing environment in which the usability assessment should be conducted [13–16]. Of the remaining 12 articles (shown in Table 2), the use of an instrumented car is recommended twice [18,19], while the additional use of a high-fidelity driving simulator is recommended by [18]. Ten of the 12 articles recommend or use a driving simulator [4–11,17,18]. The details of the simulator are specified in most of the study articles. A fix-base simulator is used in four articles [4,7,9,10], a moving-base simulator is used in two cases [5,6], and in one other case, a low-fidelity simulator is described [11]. Ref. [12] does not use an instrumented car or driving simulator; rather, desktop methods are applied where paper and video prototypes are evaluated.

**Table 2.** Aggregation of the testing environments.

| Article | Driving Simulator | Instrumented Car | Desktop Methods |
|---|---|---|---|
| Forster et al. (2019a) [4] | Fix-base | | |
| Forster et al. (2019b) [5] | Moving-base | | |
| Forster et al. (2019c) [6] | Moving-base | | |
| Forster et al. (2019d) [7] | Fix-base | | |
| Guo et al. (2019) [8] | x | | |
| Kettwich et al. (2016) [9] | Fix-base | | |
| Morgan et al. (2018) [10] | Fix-base | | |
| Naujoks et al. (2017) [11] | Low-fidelity | | |
| Richardson et al. (2018) [12] | | | Workshop |
| Naujoks et al. (2019a) [1] [17] | x | | |
| Naujoks et al. (2019b) [1] [18] | High-fidelity | x | |
| Pauzie and Orfila (2016) [1] [19] | | x | |

[1] Theoretical article.

Driving simulators are the prevalent testing environment in the field of usability assessments of ADS HMIs for CAD. Only two of the theoretical articles stress the need for real driving experiments, e.g., with instrumented cars. Driving simulators provide efficient and risk-free testing environments that provide valuable results [26]. For some research questions, they may even be the only realizable testing environment, e.g., for testing critical situations in automated driving such as near crashes and system failures in high-speed conditions. As the name implies, driving simulators do not equate with reality. High-fidelity driving simulators increase the match with reality and are to be preferred over low-fidelity simulators or desktop methods. The validity of driving simulators is assessed in several studies [27]. For research results obtained in driving simulators used to assess the usability of CAD

HMIs, the validity is yet to be verified. For practical reasons, driving simulators constitute the best testing environment. However, a validation check for the research results is needed.

### 2.2.3. Sample Characteristics

This subsection aggregates the sample characteristics. Usability tests can be conducted with experts or potential users [28,29]. Information on the participant group is provided by 14 articles of this review [4–14,16–18]. Three theoretical articles recommend including both sample groups, i.e., experts and participants in the development process of an ADS HMI [13,16,18]. Two other theoretical articles list users as participants [14,17]. Ref. [17] recommends a diverse age distribution as advised in [30]. Moreover, the authors emphasize that participants should not be affiliated with the tested system. Of the nine study articles, two conducted the usability assessment with 6 or 5–9 experts, respectively [11,12]. In these articles, experts were described as working in the "field of cognitive ergonomics" or "field of ergonomics, HMI, and function development from university and industry". The seven other study articles conducted their usability tests with potential users [4–10]. The reported sample size varies between 12 and 57. The age distribution ranges between 20 and 62, except for [10], where older adults between 47 and 88 years old were tested. Attention should be drawn to the fact that of the seven experiments with potential users, five experiments were conducted with employees of a car maker [4–8]. Table 3 shows an overview of the sample characteristics.

**Table 3.** Aggregation of the sample characteristics.

| Article | Users | Experts |
|---|---|---|
| Forster et al. (2019a) [4] | n = 24; age 20–62; BMW employees | |
| Forster et al. (2019b) [5] | n = 52; age 20–62; BMW employees | |
| Forster et al. (2019c) [6] | n = 55; age 20–62; BMW employees | |
| Forster et al. (2019d) [7] | n = 57; age 25–60; BMW employees | |
| Guo et al. (2019) [8] | n = 22; age 24–61; Renault or IRT System X employees | |
| Kettwich et al. (2016) [9] | n = 12; age 23–49 | |
| Morgan et al. (2018) [10] | n = 31; age 47–88 | |
| Naujoks et al. (2017) [11] | | n = 6; field of cognitive ergonomics |
| Richardson et al. (2018) [12] | | $n_1 = 5$, $n_2 = 9$; field of ergonomics, HMI, driver assistance systems; from university and industry |
| Forster et al. (2018) [1] [13] | x | x |
| François et al. (2016) [1] [14] | x | |
| Naujoks et al. (2018) [1] [16] | x | x |
| Naujoks et al. (2019a) [1] [17] | n > 20; diverse age distribution [30]; potential users, comparable prior experience, not affiliated with tester's company | |
| Naujoks et al. (2019b) [1] [18] | x | n > 4 |

[1] Theoretical article.

Conducting tests with potential users is the predominant method in the articles of this review. Using experts as participants represents an efficient approach for identifying major usability issues early in the development process. At advanced stages, tests with potential users are indispensable. The participants should be selected with high demands to the representativeness. The population of potential users of ADS has a high level of variability in its characteristics, e.g., prior experience or physical and cognitive abilities. User testing is most valid and productive when a sample representing potential users is being tested. Research using subpopulations could lead to biased results [31]. Therefore, when testing the usability of CAD HMIs, efforts should be made to keep the number of participants with affiliations to technical or automotive domains to a minimum. Further characteristics such as age or gender should be selected according to the represented user group. The sample size varies greatly in the selected articles. The decision on sample size should be defined by the statistical procedure used to identify potential effects of interest.

### 2.2.4. Test Cases

The test cases in an experiment are strongly dependent on the research question. As the research questions in the selected articles of this review all focus on the usability assessment of ADS HMIs for CAD, the test cases are comparable. However, no details are considered; Table 4 shows only test case categories. Ten of the 13 articles that provide information on test cases list transition scenarios [4–7,11,12,15–18]. Downward transitions are found in each of these 10 articles. A more detailed view shows that seven of these articles describe transitions to manual driving [6,7,11,12,16–18]. Eight articles [4–7,12,16–18] list test cases with upward transitions, e.g., SAE Level 0 (L0) to SAE Level 3 (L3) [1]. The system mode as well as the availability of automated driving modes are listed as dedicated test cases in four articles [12,16–18]. Likewise, three experiments include test cases with information on planned maneuvers, e.g., lane changes [7,11,12]. Two articles include test cases that represent different traffic scenarios, e.g., traffic density [8,9]. Use of the navigation function is the focus of [10].

**Table 4.** Aggregation of the test cases.

| Article | Upward Transitions [2] | Downward Transitions [2] | System Mode/Availability [2] | Others |
|---|---|---|---|---|
| Forster et al. (2019a) [4] | L0 → L2<br>L0 → L3<br>L2 → L3 | L3 → L2 | | |
| Forster et al. (2019b) [5] | L0 → L2 (driver)<br>L0 → L3 (driver)<br>L2 → L3 (driver) | L3 → L2 (driver) | | |
| Forster et al. (2019c) [6] | L0 → L2<br>L0 → L3<br>L2 → L3 | L3 → L0 L3 → L2 L2 → L0 | | |
| Forster et al. (2019d) [7] | L0 → Lx (initial)<br>L0 → Lx (re-activation)<br>L0 → Lx (re-activation) | Lx → L0 (driver) Lx → L0 (system; TOR)<br>Lx → L0 (driver; TOR) | | Maneuver (lane change, speed adaptation) |
| Guo et al. (2019) [8] | | | | Highway entry section with different traffic conditions |
| Kettwich et al. (2016) [9] | | | | Environment (traffic light) |
| Morgan et al. (2018) [10] | | | | Operating a navigation system |
| Naujoks et al. (2017) [11] | | Lx → L0 | | Maneuver and environment (splitting lanes, curvature, speed limit) |
| Richardson et al. (2018) [12] | L0 → Lx | Lx → L0 | x | |
| Gold et al. (2017) [1] [15] | | x | | |
| Naujoks et al. (2018) [1] [16] | 84 TC | 84 TC | 14 TC | |
| Naujoks et al. (2019a) [1] [17] | L2 → L3 | L3 → L2 (driver)<br>L3 → L2 (system) L3 → L1 (system)<br>L3 → L0 (system) | L2 steady state L3 steady state L3 degraded L3 unavailable | |
| Naujoks et al. (2019b) [1] [18] | L0 → Lx | Lx → L0 | x | |

[1] Theoretical article. [2] [1].

In the articles considered in this review, most of the test cases comprise transitions between or the availability of different automation modes, which are mostly referred to as SAE levels [1]. Successful transitions and the operator's understanding of the automated driving modes are important for the safe and efficient handling of the ADS. If the usability is tested and the human operator fails to understand the information communicated by the HMI, improvement measures for the HMI are inevitable. Therefore, the interaction of the operator with the ADS should be tested regarding these

functions. In addition to test cases directly related to automation modes, another type of test case can be applied when assessing the usability. These are test cases where usability evaluations refer to the handling of additional systems such as navigation systems or the radio. Non-driving-related activities (NDRA) are of high importance for usability evaluations where the human operator is involved in the driving task [2]. With the introduction of CAD, the focus of usability assessments is on transitions and the automation modes themselves. Additionally, this review concludes with a recommendation for testing non-critical scenarios. Critical situations are important for assessing safety aspects. These situations have a low probability of occurring. In particular, situations with high criticality are not suitable for usability assessments, e.g., tests that determine the range of reaction times with a crash rate of 100%. For a thorough evaluation of usability, comprising constructs such as satisfaction of the ISO Standard 9241 [23], recurring non-critical situations are more appropriate.

2.2.5. Dependent Variables

Three of the theoretical articles do not provide information on dependent variables [14–16]. The dependent variables stated in the theoretical articles or applied in the study articles of the remaining 13 articles are shown in Table 5. The dependent variables are categorized in constructs, while information on the specific metrics is added in the respective cells. More generally, the variables can be categorized into observational and subjective data. Six articles recommend or report the use of observational data [4,5,8,11,13,19]. Ref. [13] recommends collecting both data types; the interaction performance with a system or secondary task, as well as the visual behavior. Two other articles name visual behavior (e.g., the number of gaze switches) as a suitable metric [5,19]. The interaction performance is assessed either directly based on the reaction time or the number of operating steps/errors or indirectly by expert assessments. In total, four articles list this type of a dependent variable [4,8,11,13]. The SUS [20] is widely used and belongs to the subjective measures. The questionnaire is listed by six of the 13 articles [6,7,10–13]. Two other dedicated usability questionnaires are utilized in one article each; the Post-Study System Usability Questionnaire [21] by [13] and the standardized ISO 9241 Questionnaire [32], as cited by [12]. Other constructs that interrelate with usability such as acceptance, which correlates with the construct satisfaction of ISO 9241 [23], are tested by several articles in this review. These constructs report further questionnaires. Questionnaires on acceptance are used three times [7,9,19], e.g., the VDL [22] or the Unified Theory of Acceptance and Use of Technology (UTAUT) [33]. Questionnaires on trust such as the General Trust Scale (GTS) [34] or the Universal Trust in Automation scale (UTA) [35] are reported three times [7,10,19]. Constructs such as workload (cited by [10,19]), measured, for example, using the metric NASA Task Load Index (NASA-TLX) [36], situation awareness (cited by [10,19]), measured, for example, using the metric Situation Awareness Global Assessment Technique (SAGAT) [37], or the mental model of drivers (cited by [4,5]), measured, for example, using the mental model questionnaire by Beggiato [38], are each listed twice. Additional questionnaires that are reported only once can be found in Table 5. In addition to questionnaires, methods such as the Thinking Aloud Technique [39], applied by [8,9,11], or heuristic evaluations [40], applied by [12,17,18], are commonly used, especially for expert studies. Furthermore, interviews, expert evaluations, and spaces for suggestions and comments are often used to gain insights that standardized methods cannot provide [8,9,11,19].

**Table 5.** Aggregation of the dependent variables. NDRA: non-driving-related activities. UEQ: User Experience Questionnaire. meCUE: modular evaluation of key Components of User Experience. SART: Situation Awareness Rating Technique. ATCQ: Attitudes Towards Computers Questionnaire. DALI: Driving Activity Load Index.

| Article | Observational Metrics (Visual Behavior, Interaction and NDRA Performance, etc.) | Usability Questionnaire | Other Constructs (Questionnaires) and Methods |
|---|---|---|---|
| Forster et al. (2019a) [4] | Experimenter rating | | Mental model [38] |
| Forster et al. (2019b) [5] | Visual behavior (no. of gaze switches) | | Mental model [38] |
| Forster et al. (2019c) [6] | | SUS [20] | |
| Forster et al. (2019d) [7] | | SUS [20] | Acceptance (VDL [22], UTAUT [33]); trust (Trust in Automated Systems [41], UTA [35]); user experience (AttrakDiff [42], UEQ [43], meCUE [44]) |
| Guo et al. (2019) [8] | Time & frequency of button use | | Interview; Thinking Aloud Method [39] |
| Kettwich et al. (2016) [9] | | | Acceptance (VDL [22]); interview thinking aloud method [39] |
| Morgan et al. (2018) [10] | | SUS [20] | Workload (NASA-TLX [36]); Trust (ATS [41], GTS [34]); Situation Awareness (SART [45]); Technical Affiliation (ATCQ [46]) |
| Naujoks et al. (2017) [11] | Take-Over Performance No. of unnecessary system deactivations | SUS [20] | Interview; Expert Evaluation |
| Richardson et al. (2018) [12] | | SUS [20], ISO 9241 [32] as cited by [12] | Desirable HMI Aspects [47]; Thinking Aloud Method [39]; Heuristic Evaluation [40] |
| Forster et al. (2018) [1] [13] | Visual Behavior; Reaction Times; Interaction and NDRA Performance; Expert Assessment | SUS [20], PSSUQ [21] | |
| Naujoks et al. (2019a) [1] [17] | | | Heuristic Evaluation [40] |
| Naujoks et al. (2019b) [1] [18] | | | Heuristic Evaluation [40] |
| Pauzie, & Orfila (2016) [1] [19] | Visual Behavior | | Acceptance; Workload (DALI [48]); Trust; Situation Awareness (SAGAT [37], SART [45]); Interview |

[1] Theoretical article.

Summarizing the listed dependent variables, usability appears as a well-defined construct ([23]) that can be assessed via multifaceted metrics. Depending on the research questions, different dependent variables seem more applicable than others. Nevertheless, patterns can be detected. A combination of observational and subjective data is used by 6 of the 13 articles that provide information on dependent variables [4,5,8,11,13,19]. The SUS [20] is widely used by the researchers cited in this review. Where individual research questions are concerned, further questionnaires can be used to evaluate constructs such as trust, acceptance, or workload. If information for specific research interests cannot be extracted via standardized methods, interviews, the Thinking Aloud Technique or heuristic evaluations can be applied. When combining these dependent variables, mutual impacts should be considered. For example, applying the Thinking Aloud Technique is not suitable in combination with interaction performance measurements such as reaction times. For tests with potential users, this review recommends a combination of observational metrics that measure the behavior and subjective metrics that gather the operator's personal impressions. For observational data, analysis of the visual behavior based on ISO 15007 [49] and the interaction performance using the ADS HMI seem most applicable. Possible metrics are the number of operating errors, the reaction time for a button press or the percent time on an area of interest, e.g., the instrument cluster. The SUS is recommended as a

valid and widely used usability questionnaire. Supplementary questionnaires should be selected with regard to the specific research question. If usability is not the only construct of interest in an experiment, the link between the dependent variables and the constructs should be clearly stated. Standardized metrics should be used to enable comparisons between experiments and create transparency with other researchers. Short interviews provide valuable insights that can be tailored to the specific research question. Interviews should be conducted after test trials and questionnaires to avoid distorted results.

### 2.2.6. Conditions of Use

When the usability of a system is tested, the duration of use and the prior experience need to be considered. The conditions can range between the first contact between a novice user and the system and everyday use by an experienced user. The first contact can be tested with prior experience, e.g., after reading the manual, after conducting an interactive tutorial, or after being instructed by an advisor. Prolonged use can be interpreted as a series of repeat contacts between the user and the operator within a few hours or in the scope of a long-term study. The articles analyzed in this review generally do not provide detailed information on the conditions of use that is of research interest. Table 6 shows an overview with aggregated information on the nature of the usability testing provided by 14 of the articles in this review. In all 14 articles, first contact is tested or, in the case of the theoretical articles, it is recommended to be tested [4–14,16–18]. In four of these cases, the testing circumstances are specified as testing intuitive use without having first being given detailed instructions [5,6,8,16]. Another article investigates the influence of different tutorials and therefore tests both tutorials and intuitive use [4]. Of the 14 articles that test first contact, seven also assess repeat contact with the system [5,6,10,12–14,17].

**Table 6.** Aggregation of the conditions of use.

| Article | First Contact | Repeat Use |
|---|---|---|
| Forster et al. (2019a) [4] | Intuitive use, manual, and interactive tutorial | |
| Forster et al. (2019b) [5] | Intuitive use | x |
| Forster et al. (2019c) [6] | Intuitive use | x |
| Forster et al. (2019d) [7] | x | |
| Guo et al. (2019) [8] | Intuitive use | |
| Kettwich et al. (2016) [9] | x | |
| Morgan et al. (2018) [10] | x | x |
| Naujoks et al. (2017) [11] | x | |
| Richardson et al. (2018) [12] | x | x |
| Forster et al. (2018) [1] [13] | x | x |
| François et al. (2016) [1] [14] | x | x |
| Naujoks et al. (2018) [1] [16] | Intuitive use | |
| Naujoks et al. (2019a) [1] [17] | x | x |
| Naujoks et al. (2019b) [1] [18] | x | |

[1] Theoretical article.

Testing the first contact when assessing the usability of an ADS HMI appears to be the predominant method. Only a few of the selected articles tested repeat contacts when assessing usability. Prolonged use in the form of a long-term study testing everyday use is not considered in the articles selected for this review. Both first contact and prolonged use are important aspects to consider when evaluating usability. A successful first contact is highly important from the point of view of safety. This means that the handling of a system is intuitively understandable without consulting the manual—similar, for example, to a human driver using a rental car without first familiarizing themselves with the car's handling. For research effects such as disuse, misuse, or even abuse of a system, the consideration of prolonged use in everyday situations is critical [50]. As it poses a different type of research question, it requires a different kind of experiment. In alignment with most of the articles, this review concludes by recommending first contact tests. The NHTSA minimum requirements state that an HMI must be designed in such a way that a user understands if the ADS is "(1) functioning properly; (2) currently

engaged in ADS mode; (3) currently "unavailable" for use; (4) experiencing a malfunction; and/or (5) requesting control transition from the ADS to the operator" [24] (p. 10). The fulfillment of these requirements can be checked by assessing usability in a first contact situation. This requires that participants are not given detailed instructions, such as pictures of the HMI requesting a control transition, prior to the first contact. Instead, participants should only receive instructions with general information on ADS.

## 3. Discussions

In this review paper, 16 selected articles focusing on usability assessments for ADS HMIs for CAD are analyzed. Information on methodological approaches, study characteristics, as well as the understanding of the term usability has been aggregated. The insights gained are used to draw conclusions on best practice for researchers investigating the usability of CAD HMIs. In this section, the recommendations are discussed and incorporated in more general advice on usability testing.

Three different sources are cited for the understanding of the term usability [23–25]. Yet, many articles do not provide information on the definition used. The definitions of the three sources result in different study designs than those that would have been derived had a different definition been selected. In order to assess the usability of CAD HMIs, we advise applying a combination of ISO 9241 and the NHTSA minimum requirements [23,24]. However, other definitions, e.g., [25], might be better suited for specific research questions. In general, it is important to provide an operationalization of the term usability when conducting assessments, especially where standards are not applied.

For practical reasons, the review concludes with the recommendation of high-fidelity driving simulators. Depending on the development stage, other testing environments may prove more applicable. For early prototypes, desktop methods provide valuable insights with minimal resource input. Real driving tests can help in the refinement process of preproduction products.

This review recommends that usability tests should be conducted with potential end users. These tests are indispensable for final usability evaluations. Other participants, such as experts or users that represent only a segment of the user population, e.g., students or participants with affiliations to technical or automotive domains, can provide valuable insights at earlier stages of the development process.

The test cases listed in the best practice advice of this review focus on transitions between, and the availability of, different automation levels in non-critical situations. These test cases are recommended for general usability assessments of ADS HMIs for CAD. Other test cases in this review cover usability assessments of HMIs displaying information on more complex scenarios, such as maneuvers, navigation systems, or dense traffic. These test cases are relevant for specific research questions, e.g., the design of integrated functions in the CAD HMI.

A set of metrics for testing the usability of CAD HMIs is listed in this paper. Depending on the study design and the research question, further metrics might prove suitable in obtaining valuable research findings. Researchers should clearly indicate the link between dependent variables and the respective definition or construct of interest.

This review recommends that the usability tests be performed in first contact situations without in-depth instructions on how to use the system having been provided prior to the testing situation. Where research questions not focusing on the NHTSA minimum requirements are concerned, the use of manuals or tutorials might be applicable in order to equalize the knowledge and experience level of the participants. In addition to testing first contacts, the everyday use of ADS is of great interest, especially in the context of CAD. The transition of the human driver from operator to passenger could generate side effects such as disuse, misuse, or abuse of the ADS, which might impair safety. The assessment of these effects poses an interesting and important topic for future research.

## 4. Conclusions

This paper reviews 16 articles, comprising both study and theoretical articles. These articles are analyzed in respect of six study characteristics. The insights into common practice and theoretical considerations lead to a derivation of best practice advice. This advice is aimed at helping researchers who are interested in usability assessments of CAD HMIs in the planning phase of a study. Furthermore, the comparability of studies in this field increases with the application of similar experimental designs. Table 7 summarizes the key statements of the derived best practice.

**Table 7.** Best practice advice for testing the usability of conditionally automated driving (CAD) human–machine interfaces (HMIs). ADS: automated driving system.

| Study Characteristic | Best Practice Advice |
| --- | --- |
| Definition of Usability | General Definition: "extent to which a system, product or service can be used by specified users to achieve specified goals with effectiveness, efficiency and satisfaction in a specified context of use" [23] (p. 2)<br>Practical Realization: the user understands that the ADS is "(1) functioning properly; (2) currently engaged in ADS mode; (3) currently "unavailable" for use; (4) experiencing a malfunction; and/or (5) requesting control transition from the ADS to the operator" [24] (p. 10) |
| Testing Environment | Driving Simulator |
| Sample Characteristics | Sample Group: represents the potential user population (age, gender, prior experience, affiliation with technical devices, etc.)<br>Sample Size: determined by the statistical procedure |
| Test Cases | Scenarios: (1) transitions between different automation modes and (2) availability of different automation modes<br>Criticality: non-critical situations |
| Dependent Variables | General: Combination of observational and subjective metrics<br>Observational metrics: (1) visual behavior according to [49] (e.g., percent on Area of Interest) and (2) the interaction performance with CAD HMI (e.g., operating errors or reaction time for a button press)<br>Subjective Metrics: (1) System Usability Scale [20], (2) short interviews after test trials and questionnaires, and (3) supplementary standardized questionnaires |
| Conditions of Use | First contact between user and ADS<br>Instructions contain only general information on the ADS |

## 5. Outlook

In this review, driving simulators are identified as the prevalent testing environment in the field of usability assessments of ADS HMIs for CAD. As an efficient and risk-free alternative to real driving experiments, simulators offer a convenient and valuable testing environment. Since the validity of driving simulators has not yet been assessed, the transferability of results to the real world is not assured. A thorough validation study comparing a simulator and a test track experiment is advisable. This forms the foundation for a future validation study series comprising a driving simulator experiment and three real driving experiments on test tracks in Germany, the USA, and Japan.

**Author Contributions:** Conceptualization, D.A., J.R., S.H., F.N., A.K., and K.B.; Methodology, D.A., J.R., A.L., S.H., and F.N.; Formal Analysis, D.A., J.R., A.L., and S.H.; Writing—Original Draft, D.A.; Writing—Review and Editing, D.A., J.R., A.L., S.H., F.N., A.K., and K.B.; Supervision, A.K., and K.B. All authors have read and agreed to the published version of the manuscript.

**Funding:** This research was funded by the BMW Group.

**Conflicts of Interest:** The authors declare no conflict of interest. The funders had no role in the collection, analyses, or interpretation of the data.

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
