# Peer review of "Usability Evaluation—Advances in Experimental Design in the Context of Automated Driving Human–Machine Interfaces"

_information, doi:10.3390/info11050240_

Round 1
Reviewer 1 Report
Human computer interface evaluation in automated driving is an important research area, and automated driving will be ubiquitous in our everyday lives. The review paper is informative and technically sound. I do not see potential issues.
Reviewer 2 Report
The paper gives an overview of the state of the art regarding usability assessment of HMIs in the context of autonomous driving. The Authors performed a literature review based on a specific guideline in order to identify a number of relevant papers to use as a reference for their investigation.
The Authors identify six features to consider as metrics for evaluating the usability of an HMI: Definition of Usability, Testing Environment, Sample Characteristics, Test Cases, Dependent Variables and Conditions of Use. For each of them, a summary of what the selected researchers pursued in their works is provided, and recommendations for best practices to use in future research and development activities are given with the aim of building a methodological approach.
I found the work interesting and I agree on the need to define standards in a rather new field. This manuscript provides some indications but cannot be considered a comprehensive work. As the Authors stated, for instance, regarding the test environment: driving simulators have been identified as the main test facility, however, their validity has not yet been assessed and the transferability of the results to the real world is not guaranteed. It would be interesting to read about the next steps of this research and conclusions based on measurable data.